# Prediction System for Overhead Cranes Based on Digital Twin Technology

**Pei Gao** [1,2] **, Zongyan Wang** [1,*]**, Yuting Zhang** [1] **and Menglong Li** [1]

1   School of Mechanical Engineering, North University of China, Taiyuan 030051, China
2   Department of Mechanical and Electrical Engineering, Shanxi Polytechnic College, Taiyuan 030006, China
*   Correspondence: wangzongyan@nuc.edu.cn; Tel.: +86-139-3422-1090

**Abstract:** To solve the problems of inaccurate quality inspection and poor safety maintenance of traditional overhead cranes, this study has developed a prediction system for overhead cranes based on digital twin technology. First, interworking of the data flow between the control port of the overhead crane and the digital twin system is realized. Then, finite element technology is exploited to calculate the stress of the crane bridge, thereby performing stress rendering of the digital twin system. Next, the life of bridge cranes is predicted by the stress and strain of their main girders. Finally, synchronous overhead-crane communication combines the virtual twin system and the actual control system of the crane. The study results provide a reference for developing crane software based on digital twins.

**Keywords:** digital twin; cranes; data flow; stress rendering; virtual twin system



## 1. Introduction

Digital twin technology is used to map the real world to the virtual world through perception and calculation to realize virtual–real fusion and cognitive prediction. It is an important approach to realize "simulation" [1]. Application of digital twin technology to predictive maintenance of overhead cranes requires building an accurate model of an overhead crane and monitoring and tracking it. The working status of each component of the overhead crane can be fully grasped through formulating corresponding standards. This method can detect the operating status of the overhead crane and predict the service life of the main components, so it is significant for the normal operation and safety precautions of overhead cranes [2]. Digital twin simulation is to keep in sync with its reflected data of the overhead crane in terms of appearance, content, and movement speed, and the state will be the same [3]. Though the simulated crane is a virtual 3D software product, it can truly represent the performance of the physical body. In actual experimental operations, it can remotely monitor the running status of a crane from a safe distance and issue work instructions such as start, stop, forward, and backward. Traditionally, the operating experience of machinery and equipment is vague, difficult to grasp, and indispensable for judgment [4]. In contrast, application of simulation digital twin technology can digitally manage experience that could not be saved before, as well as save, modify, copy, and migrate the experience [5]. Based on currently achieved program simulation, the thinking imagination can be further broadened. Meanwhile, a complete database of crane operating parameters can be established through repeated experiments to achieve data collection, storage, and analysis. In addition, through using new-generation technologies such as artificial intelligence, big data, and edge computing, all the states in each step of the overhead crane can be predicted, and evaluation and warnings of its potential risks can be made in advance [6].

The rest of this paper is organized as follows. Section 2 describes the research progress of advanced digital twin technology. Section 3 introduces the design of the basic digital

twin software. In Section 4, the remaining service life of a bridge crane is predicted using an algorithm. In Section 5, the simulation experiment and the virtual–real synchronization experiment of the digital twin for the crane are conducted. Section 6 makes a technical summary of the crane digital twin software.

The contributions of this paper are as follows:

1. The digital twin model is established to read the real-time lifting operation information of the physical overhead crane, realize synchronous mapping of the lifting operation scenario of the physical overhead crane in the virtual environment, and simulate lifting operation tasks that have been set to verify the feasibility of lifting operations.
2. To monitor the physical bridge-crane-lifting operation, the virtual bridge crane can synchronize and monitor the lifting operation of the physical bridge crane in real time, thus assisting the physical bridge crane to achieve predictive maintenance.
3. A good human–computer interaction interface is provided, making it easy for the operator to give control instructions to the digital twin model and obtain the bridge crane's real-time operational data.
4. The life of the overhead crane is predicted using the stress and strain of the main beam of the overhead crane.

## 2. Literature Review

### 2.1. Serial Communication

With the development of microcomputer systems and the Internet of Things (IoT), the information exchange between computers, as well as between computers and other external hardware devices, is becoming increasingly important [7]. Serial communication is a communication method that transmits data bit by bit between an external hardware device and a computer. This process usually involves multiple interface protocols, such as data formatting, serial-to-parallel data conversion, and TTL (time to live)- and EIA-level (embedded logic analyzer) conversion. The entire communication process is realized using data signal lines, ground wires, and control lines [8]. In a distributed control system composed of multiple microcomputer processors and external devices, attributed to the natural advantages of serial communication for bit-by-bit data transmission, fewer data lines are needed, thus realizing low-cost, long-distance information transmission. However, in practical applications, the serial port method is widely used to control real-time, short-distance, and simple external devices such as terminals, printers, controllers, etc. [9].

### 2.2. Digital Twin Technology

Digital twins are defined as highly integrated multiscale, multiphysics, and multi-probability simulation models that fully utilize sensors, physical models, and historical data to reflect the full life cycles of physical products in real time [10]. After the concept of the digital twin was proposed, many scholars built mapping models based on digital twin technology to solve system problems in various fields. This makes the basic theory of digital twin technology increasingly perfect, showing great potential in the fields of planning and design, manufacturing, and fire protection [11]. For example, Zhao et al. [12] proposed a prediction method for residual service life through combining digital twin technology with transfer learning theory and an embedded convolutional long–short-term memory expansion model for actual fault diagnosis and main bearing life prediction. Tao et al. [13] presented emerging digital twin technology for complex equipment to realize physical–virtual fusion; meanwhile, they constructed a general digital model of complex equipment and proposed a new prediction and health management method driven by digital twins, thus effectively utilizing the interaction mechanism and fusion data of digital twins. Wang et al. [14] incorporated digital twins into the existing data-driven intelligent customization paradigm to make this process more responsive, adaptable, and predictive, and then they proposed a new framework for data-driven intelligent customization enhanced by digital twins. Liu et al. [15] designed a real-time workshop digital twin scheduling platform for discrete manufacturing. It can improve the flexibility

of an intelligent workshop and response processing speed after dynamic disturbances; also, it can monitor the physical workshop in real time and track information such as orders, products, and equipment.

### 2.3. Life Prediction for Overhead Cranes

A bridge crane is typical mechanical equipment, and the service life of its components will be affected by various factors, including quality of raw materials, the installation process of the components, level of daily operation, maintenance of the equipment, etc. Therefore, in life prediction for bridge-crane components, it is necessary to clarify the principles of the roles of various influencing factors, thus establishing a more accurate life-prediction-calculation formula [16]. Through recording and analyzing the production and operation parameters of equipment, the parameters affecting the operating life of the equipment are integrated, and the remaining service life of the equipment is predicted using scientific analysis methods [17]. When there are sufficient equipment production and operation data, the remaining service life of the equipment can be predicted using intelligent algorithms [18]. Dong Qin et al. proposed a real-time fatigue-life-prediction method for bridge structures based on digital twins. The specific type of generic bridge crane was selected as the physical entity of the research object, and the information collection system was applied to obtain the current service-state information of the physical entity [19]. A fatigue-life-assessment method was proposed by Xu Bin et al. The finite element method and experimental stress analysis were employed to evaluate the service condition of a ship-unloading crane according to the relevant criteria and the nature of the risk [20].

## 3. Development of the Digital Twin System for Overhead Cranes

### 3.1. Framework Design

Unity3D was originally proposed as a 3D game development tool. With the development of the industrial IoT, it has been widely used in multiple digital twin simulation fields. It can not only render 3D simulation models in real time but also display received sensor data or cloud platform data synchronously. With this tool, staff can grasp actual usage status information for external IoT equipment on time and can use multiple platforms such as AR/VR/MR for human–computer interaction to realize remote and safe control of machinery and equipment [21].

Timely information exchange and synchronization between IoT equipment and software programs should be supported as perfectly as possible. TCP is a traditional communication protocol, and with this protocol, IoT equipment reads relevant sensor data through PLC and uploads it to the cloud server in JSON format. The front end of the Unity3D digital twin performs real-time data acquisition and processing through Socket or HTTP protocols and displays the real operating status of the IoT equipment. Similarly, peripherals can be reversely controlled through the simulation software; instructions are sent from the program to the cloud server and then forwarded to the PLC of the IoT equipment, thereby issuing device physical-operation instructions [22].

The design of the digital twin software is shown in Figure 1. Accuracy, real time, and effectiveness of data transmission are critical to ensuring the complete digital visualization process. For further design, the visualized data is highly dependent on data and information collected by sensors. In terms of software development, advanced, high-efficiency algorithms are employed to conduct global perception and operation monitoring of data. Furthermore, historically accumulated data is integrated for calculation and presentation. Moreover, IoT equipment-failure risk-assessment predictions can be made effectively via adding physical characteristics and behavioral logic calculations to the model [23].

### 3.2. Digital Twin Technology for Overhead Cranes

#### 3.2.1. Digital Twin Software Design

Software design should follow the characteristics of easy application and high simplicity, scalability, and maintainability. Common digital twin requirements are composed of

IoT equipment terminals, cloud servers, and software simulations. This experiment did not involve cloud storage or forwarding; instead, the serial communication between the crane and the simulation software was directly conducted through the com terminal transfer method. The information interaction process is illustrated in Figure 2.

## The schematic diagram of using digital twins in Unity3d

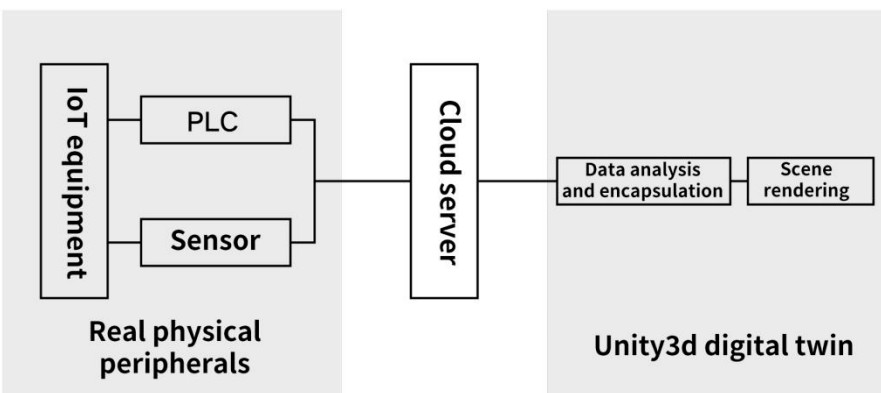

**Figure 1.** Design of the digital twin software.

## Analysis of the digital twin demand of the overhead crane

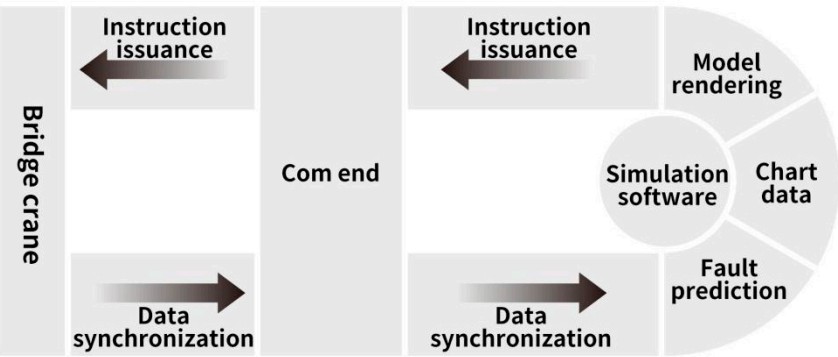

**Figure 2.** The data exchange of the overhead crane.

Considering the actual requirements of the predictive maintenance system for the overhead crane, visualization software should be able to display the stress values endured by the crane on the model in real time. Meanwhile, the stress values can be visually presented in the form of dashboard charts, so they can be used for reference analysis of the fault code when the equipment is running abnormally. Additionally, a function module is required to issue operating instructions to the crane [24]. In terms of software framework design, combined with the MVVM (model–view–viewmodel) mode, the UI module and the data communication module are packaged separately.

Program packaging should follow the characteristics of high cohesion and low coupling, and it should be implemented in a manner of object-oriented and interface programming. In addition, it should optimize serial communication efficiency to improve scene rendering quality. The program should use an asynchronous thread algorithm to analyze crane status data and make predictions and judgments. According to the above analysis, this paper formulates a preliminary software prototype design for the digital twin visualization of the overhead crane, and its effect is shown in Figure 3.

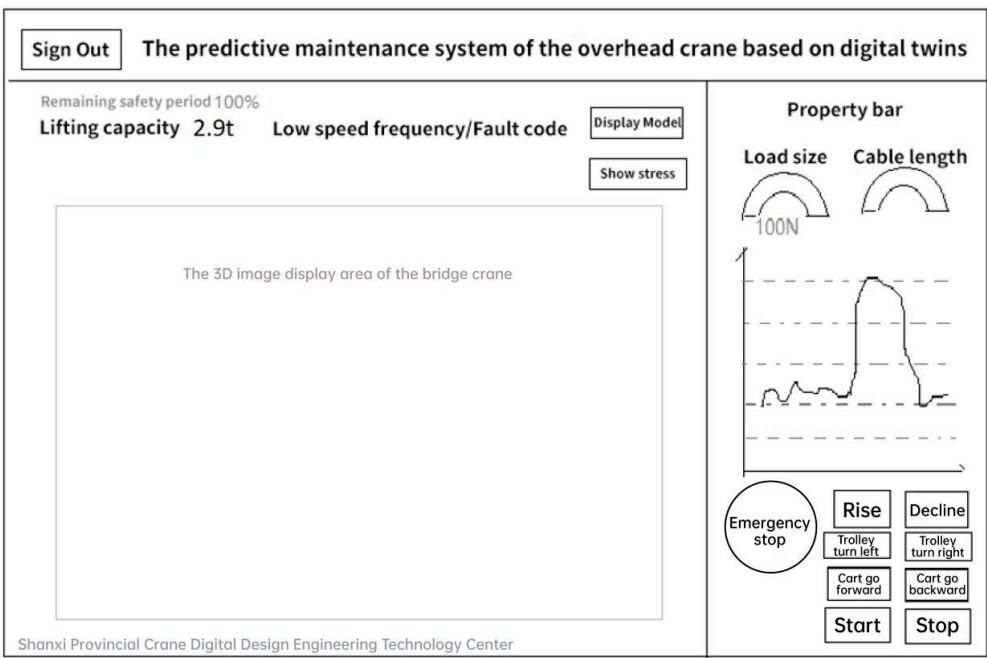

**Figure 3.** The effect of the preliminary software prototype.

### 3.2.2. Digital Twin Program Realization

The digital twin simulation software was developed based on Unity3D and Visual Studio, which were downloaded from the official website before installation and running. A new Unity3D project called CraneDigital was created first. Encapsulation of the software framework is the foundation of an excellent program, and quality directly affects the execution efficiency of the program [25]. In the previous design, it was mentioned, the UI-drawing module and data communication module were packaged independently. These two modules are the main body of the large function division of the simulation program. To further divide the frame details, the UI-drawing module can be split into the crane-model stress-graph-rendering function and the UI function; the data communication module can be split into subfunctions such as data analysis, fault prediction, and serial communication.

Folders, such as "Scenes", "Resources", and "Script", were created to store different functional modules. The created directory structure is presented in Figure 4.

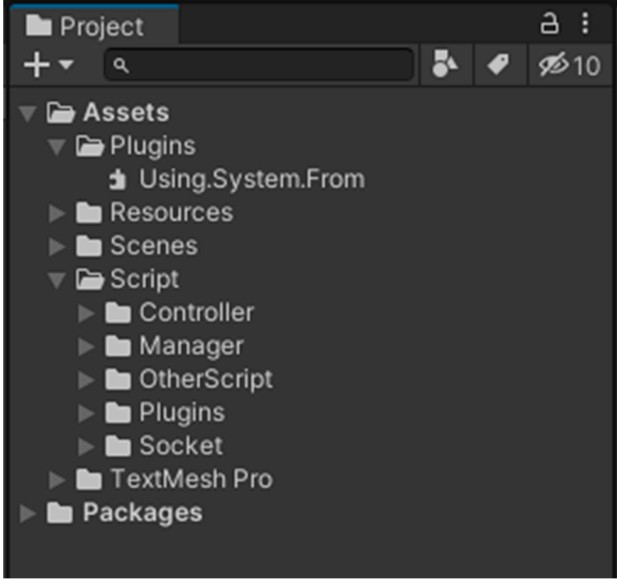

**Figure 4.** The created directory structure.

The script was named CommandUtils.cs, and it was transplanted to the Script folder of the simulation project. Then, a new scene (Scene) was created, and the camera position was adjusted to complete the framework design and preparation for development.

### 3.3. Stress Graph Development of the Overhead-Crane Model

The built overhead-crane model was dragged to the Scene. Meanwhile, two scripts were added to the model.

QZJOBJmanager.cs was used to obtain the subobjects (GameObject) of the crane model, including the crane's cart, trolley, heavy objects, and main beam. When simulating the crane lifting heavy objects, the software model could follow the movement and perform animation control. In Unity3D, a transform represents the position of a GameObject; controlling the mobile animation would change the coordinate values corresponding to its transform. The motion control of the overhead-crane model and the change in the stress graph value were monitored with serial communication. The new data received could be rendered in real time.

### 3.4. Development of the UI of the Simulation Software

In the previous requirement analysis, a preliminary design of the digital twin UI is given. The basic panel interfaces of Unity3D, such as text, buttons, and pictures, are generally realized by the corresponding Text, Button, and empty GameObject mappings. Meanwhile, the parameters, including size, position, and color value, can be adjusted through the Inspector component properties.

The key to the UI is the specific numerical display of the crane in the property bar on the right. The dashboard realizes scale drawing and data filling of the load size and cable length. In the previous section, the stress value is fed back for the rendering of the crane model, and the value is marked by color. However, the stress-trend chart is particularly important to intuitively display the fluctuation of a specific value. This study used the LineChart component in Unity3D to realize chart data drawing and bound the script LineChartController.cs to it. The main function of the new script was to add and delete the stress value data for the chart component to ensure that the chart curve could be refreshed in time. The core code for adding new stress values is presented below.

Figure 5 shows the development of the stress module for detecting the stress of the girder of the overhead crane. The lower right corner of the UI is the module used to issue instructions to the crane. Here, the key is the processing of the button-click event, which can send the operation instruction to the Com terminal in time. To optimize the program logic, the protection and prompt mechanisms for the instruction communication exception are also necessary.

### 3.5. Development of Fault-Code Prediction

An important requirement of the digital twin program is to analyze the abnormal fault code of the crane and prompt the risk. Generally, model risk is evaluated based on comprehensive analysis results of big data and rule experience. Large data volume is a notable feature of general evaluation and prediction. Therefore, in the development process, this study encapsulated fault analysis and prediction into an independent module and adopted asynchronous thread processing to ensure that the time-consuming analysis would not cause delays in the UI refreshing of the main software interface. The asynchronous thread fed back the analysis and processing results to the main thread for UI risk prompts.

To this end, a new fault-exception auxiliary script, ErrorController.cs, was created. All abnormal information during the crane operation would be recorded in the collection cache of this class. In the serial communication process, asynchronous data monitoring was maintained, and each piece of data was checked for abnormalities. Some of the key code is illustrated as follows.

The fault-exception module should have ensured the execution or waiting state in the entire running process of the program. The prompt effect of a certain abnormal message monitored through the actual experimental test is illustrated in Figure 6.

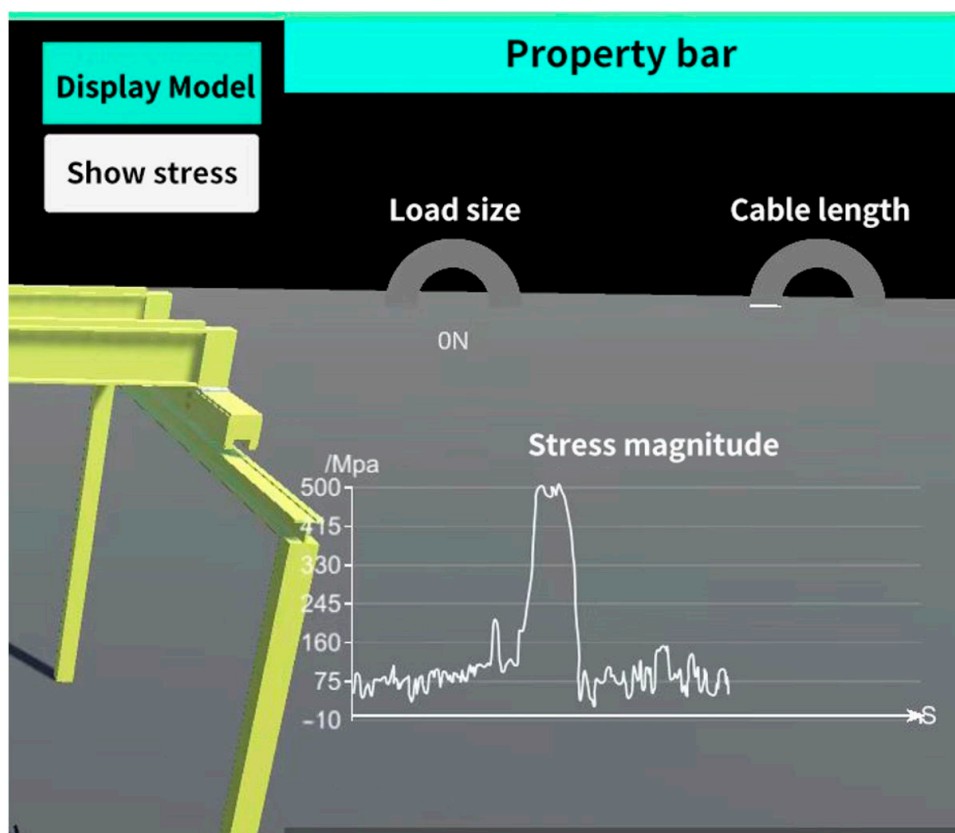

**Figure 5.** The development of the stress module.

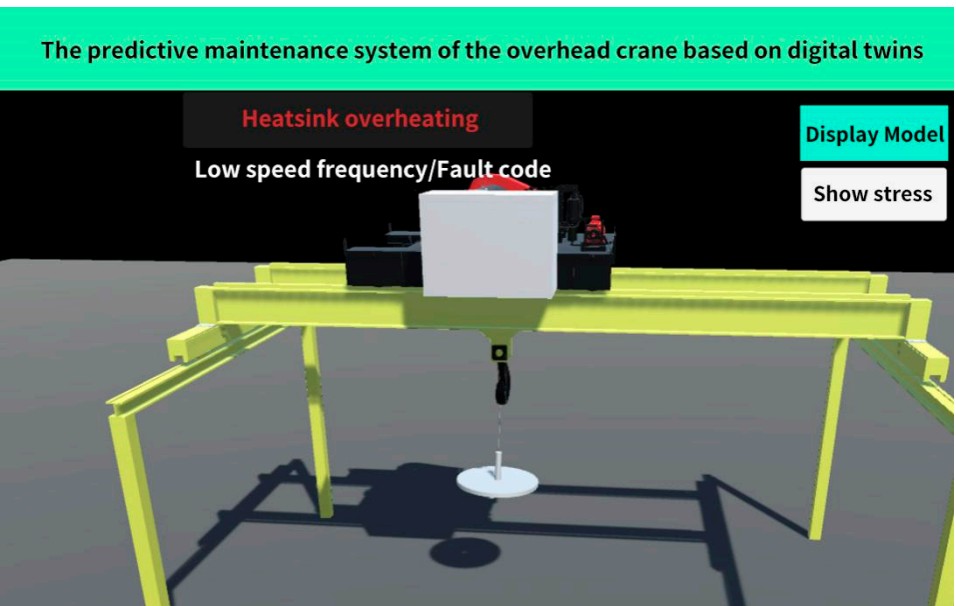

**Figure 6.** The prompt effect of an abnormal message.

### 3.6. Verification of Serial Communication Function of the Overhead Crane

In the serial communication of the overhead-crane project, the com terminal was plugged into the microcomputer system to forward and verify the cyclic redundancy

code CRC16. During this experiment, eight instructions needed to be issued to the crane, including front, back, left, right, rise, decline, start, and stop. The work instructions of the overhead crane were generally composed of eight bytes. For a specific instruction, CRC16 calculation was first conducted through a software program to obtain a two-byte check code. Then, the eight bytes of the original instruction and the two-byte check code were spliced into a new data stream of ten bytes, which were sent to the Com terminal through serial communication. After the Com terminal confirmed the verification, the corresponding work instruction was finally sent to the overhead crane.

Corresponding instruction values could be obtained from the equipment operation manual of the overhead crane. Meanwhile, the eight instruction values required for the final experiment were calculated and obtained according to the CRC16 check. Each instruction had ten bytes, and the last two bits are the check code.

Start instruction: 01 10 06 04 02 00 00 00 60 18
Stop instruction: 01 10 06 04 01 00 00 00 60 5C
Cart forward instruction: 01 10 06 04 20 00 00 00 6A 60
Cart vehicle backward instruction: 01 10 06 04 10 00 00 00 65 60
Trolley left-turning instruction: 01 10 06 04 04 00 00 00 60 90
Trolley right-turning instruction: 01 10 06 04 08 00 00 00 63 C0
Rise instruction: 01 10 06 04 80 00 00 00 48 60
Decline instruction: 01 10 06 04 40 00 00 00 74 60

The realization of the digital twin target of the overhead crane was developed based on the C# language in Unity3d. Then, the C# language and the Winform framework were exploited to assist in the development of the joint debugging serial communication function and encapsulate it into a module tool class. According to the requirements of this experiment, the serial communication tool class should have included encapsulation of basic functions such as serial port opening, serial port closing, instruction sending and receiving, etc.

The first step of serial communication is to open the serial port. The overhead crane adopted COM3 as the serial number, and the baud rate was 19,200. The key code to open the serial port link is listed below.

To facilitate the observation and demonstration, the UI of the verification program was established using the Winform framework. The final running effect of the software is demonstrated in Figure 7.

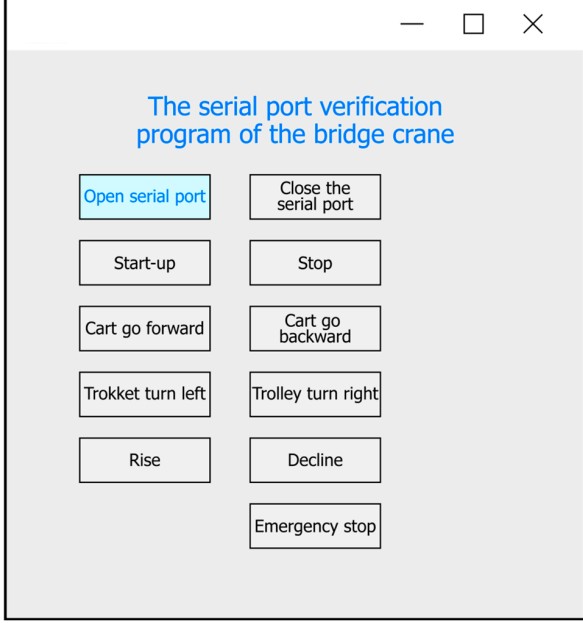

**Figure 7.** The final running effect of the software.

## 4. Life Prediction for the Bridge Crane Based on Tent-SSA-BP

### 4.1. Chaotic Sequence

In this study, the BP neural network optimized by the Tent chaotic sequence was adopted to realize the life prediction for the bridge crane based on its stress analysis. Chaos is a nonlinear phenomenon, ubiquitous in nature. Due to the characteristics of randomness, traversal, and regularity, chaotic variables are used by many scholars to optimize search problems. It can not only maintain the diversity of a population but also facilitate an algorithm to jump out of the local optimum, thereby improving the global search ability [26]. The generation steps of the Tent chaotic sequence are as follows.

1. Randomly generate the initial value, $z_0$, within (0, 1).
2. Perform the iteration using Equation (1), thereby generating the $Z$ sequence.

$$z_{i+1} = (2z_i)mod1 + rand(0,1) * \frac{1}{N_T} \tag{1}$$

3. When the number of iterations reaches the maximum, the program will terminate and save the generated $Z$ sequence.

### 4.2. Sparrow Search Algorithm

The sparrow search algorithm (SSA) is an emerging metaheuristic algorithm proposed in 2020. It belongs to the swarm intelligence algorithm based on group-based optimization of social characteristics, and it is equivalent to the particle-swarm-optimization algorithm and the dragonfly-optimization algorithm. This algorithm simulates sparrow foraging and antipredation behaviors via continuously updating individual positions [27]. Compared with traditional algorithms, the SSA has a simple structure, is easy to implement, involves fewer control parameters, and has a stronger local search ability. This algorithm performs better than traditional algorithms, such as the particle swarm algorithm and the ant colony algorithm, on unimodal and multimodal benchmark functions [28].

In the SSA, individuals are categorized into discoverers, followers, and vigilantes, and each individual position corresponds to a solution. According to the algorithm setting, vigilantes account for 10% to 20% of the population; in contrast, discoverers and followers are dynamically changing, i.e., one individual becoming a discoverer necessarily indicates that another individual will become a follower [29]. According to the labor division, the discoverer mainly provides the foraging direction and area for the entire population; the follower follows the discoverer to forage; and the vigilante is responsible for monitoring the foraging area. In the foraging process, acquisition of resources is completed through continuously updating the positions of the three types of individuals [30].

### 4.3. BP Optimized by the Chaotic Sparrow Algorithm

The flow chart of the chaotic sparrow algorithm is shown in Figure 8.

The steps of the chaotic sparrow algorithm are listed below.

1. Initialization. According to the input matrix, determine the BP topology and initialize the maximum number of iterations, $T_0$.
2. Set the population size, $N$; the number of discoverers, $p_{num}$; the number for reconnaissance and warnings, $s_{num}$; the dimension of the objective function, $D$; the upper and lower bounds of the initial value, lb and ub, respectively; the maximum number of iterations, $T$; and the solution precision, $t_0$.
3. Apply Tent to generate N D-dimensional vectors, $Z_i$, and set them to the value range of the variables through Equation (2), where $d_{min}$ and $d_{max}$ denote the minimum and maximum values of the $d$-dimensional vector, $X_{new}^d$, respectively.

$$X_{new}^d = d_{\min} + (d_{\max} - d_{\min})Z_i \tag{2}$$

4. Calculate the fitness, $f_i$. Select the best fitness, $f_g$; the worst fitness, $f_w$; and their corresponding positions, *xb* and *xw*, respectively.
5. Select the first $p_{num}$ with the best fitness as the discoverer and the rest as joiners. Then, update the positions of the discoverers and the joiners.
6. Randomly select $s_{num}$ as vigilantes and update their positions.
7. Conduct an iteration to calculate the fitness, $f_i$, and average fitness, $f_{avg}$, of each one.
8. If $f_i \geq f_{avg}$, perform chaotic perturbation on the individual. If the performance of the individual is better after the perturbation, replace the previous individual; otherwise, it will remain unchanged. If $f_i < f_{avg}$, use Equation (3) to perform Gaussian mutation on the individual, where x represents the original parameter value, $N(0, 1)$ represents the random number for normal distribution, and mutation(x) denotes the value after Gaussian mutation.

$$mutation(x) = x(1 + N(0, 1)) \tag{3}$$

9. Update the best position, *xb*, and its fitness for the entire population, as well as the worst position, *xw*, and its fitness.
10. Judge whether the maximum number of iterations or solution precision was reached. If so, the iteration will terminate, and the optimal parameter results will be output.
11. After the optimization process of the Tent-SSA algorithm is completed, the optimal parameters will be output and assigned to the BP network for prediction. Moreover, the coefficient of determination, $\varepsilon$, will be calculated.

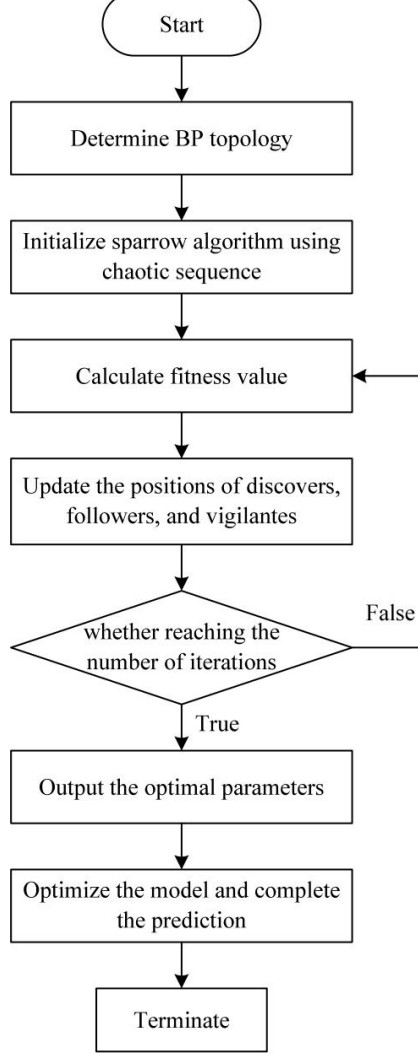

**Figure 8.** The flow chart of the chaotic sparrow algorithm.

### 4.4. Life Prediction and Analysis

Through recording and analyzing production and operation parameters of overhead cranes, the parameters affecting the operating life of the equipment were integrated, and the remaining service life of the equipment was predicted using scientific analysis methods. With sufficient equipment production and operation data, the existing data could be used for mathematical modeling to achieve the fuzzy operation of the remaining service life of the equipment; when equipment production and operation data are sufficient, the algorithm can be used to estimate the remaining service life of equipment.

Fatigue failure of the main girder of the bridge crane is the main failure mode. Thus, life prediction for bridge cranes mainly aims at the fatigue-life prediction for the main girder. The main factors affecting the fatigue life of mechanical equipment include the working environment and conditions, the stress state of the parts, and the material properties of the parts. Meanwhile, the cumulative fatigue damage theory was relatively complete. Due to the differences in the work tasks of bridge cranes, the load of the main girder was a random parameter, and the historical data of random load changes became the fatigue-life data of the main girder of the bridge crane, which was the premise of fatigue-life prediction and could reflect the change of load over time. Tent-SSA-BP was established through data acquisition and processing. Specifically, preprocessing was conducted on the stress–time curve of the main girder, and then a statistical analysis and average stress correction were performed.

## 5. Experimental Section

### 5.1. Experiment Preparation

Figure 9 presents the constructed platform for the data exchange between the crane and the computer-side digital twin system in the actual environment. The digital twin system consisted of two parts. One part is the overhead crane in the actual environment. The model was a E050-015-022N Aolitong overhead crane, with a lifting capacity of 2.9 t. The upper end of the yellow main girder was equipped with a trolley, and the upper-end control box was a PLC control system. The other part is a digital twin visualization platform built on a virtual computer platform under the Unity environment, and it was used to simultaneously detect the real-time predictive conditions of the crane. The computer was equipped with an AMD-R5-4600H CPU and an Nvidia RTX3060 GPU (12 GB memory). The digital twin virtual platform was configured using Python 3.7 and Microsoft Visual Studio 2019 C# as the development languages.

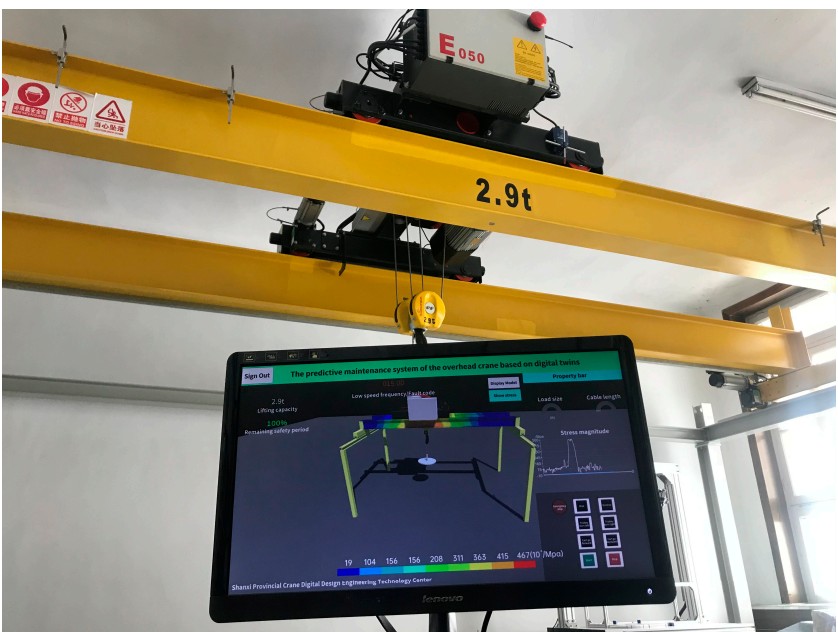

**Figure 9.** The digital twin experiment platform.

### 5.2. Experimental Analysis of the Digital Twins of the Overhead Crane

This experiment exploited data interoperability, relying on the IoT gateway and the com terminal transmitter. Each sensor of the overhead crane collected data through the IoT gateway and sent the data to the simulation software to monitor the operating parameters. The simulation software issued commands through the com terminal transmitter to each drive module of the overhead crane to achieve operational control. The location of the IoT gateway box in the bridge-crane distribution box is shown in Figure 10a. The com terminal transmitter for the digital twin system controlled the crane's physical action through sending wireless signals to the transmitter–receiver board. Then, the control board received the signal to control the crane action, as shown in Figure 10b.

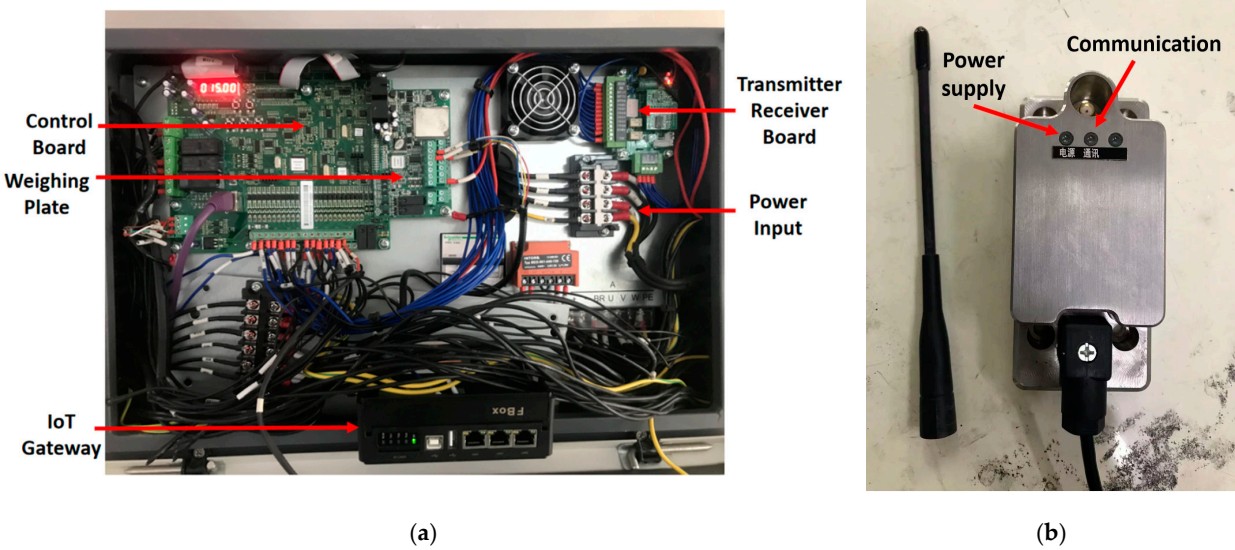

(**a**)  (**b**)

**Figure 10.** The IoT gateway communication settings. (**a**) The location of the IoT gateway box in the bridge-crane distribution box. (**b**) The com terminal transmitter.

After many instances of joint debugging as well as modification and optimization with the serial communication data of the Com port, the finalized software had an almost consistent running effect with the previous prototype design and could meet the functional requirements of the overhead-crane prediction system. Figure 11 demonstrates the running effect of the completed simulation software program.

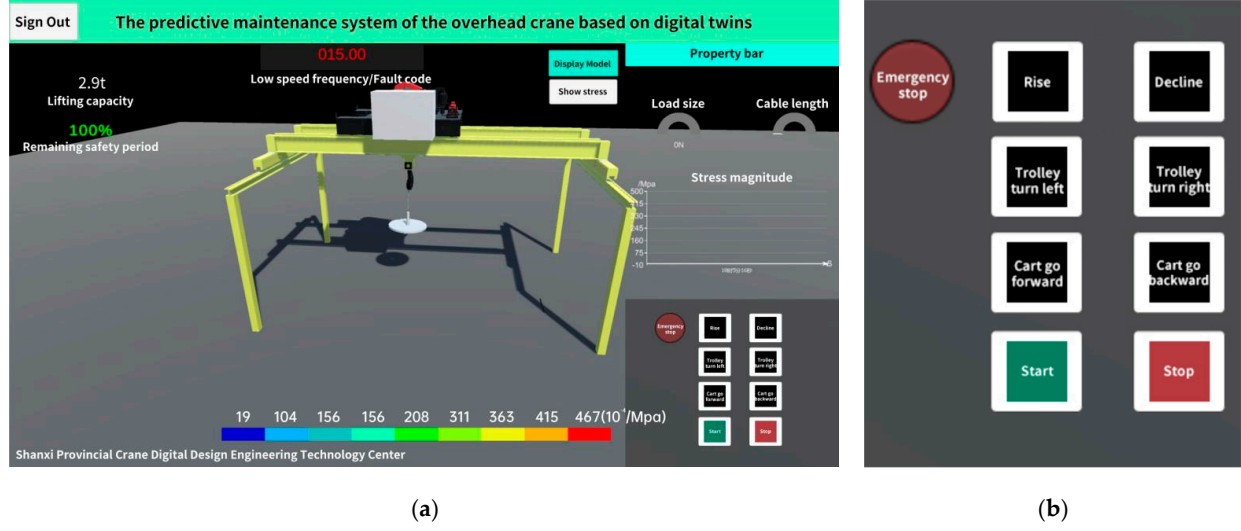

(**a**)  (**b**)

**Figure 11.** The running effect of the completed simulation software program. (**a**) Main interface. (**b**) Control interface.

### 5.3. Experimental Analysis of Life Prediction for the Overhead Crane

The fatigue failure of the main beam of the overhead crane was the most important form of failure. Thus, the life prediction for the overhead-crane components was mainly the fatigue-life prediction for the main beam. The stress–strain gauges were arranged separately for the main beam of the bridge crane in the middle position, as shown in Figure 12. The collected stress data was transmitted back to the digital twin system through the IoT gateway for analysis. The collected stress changed with time, and the digital twin interface was controlled to make the crane hook move up and down. The stress changes are shown in Figure 13a. The collected data were subjected to finite element analysis, and the analysis results are shown in Figure 13b.

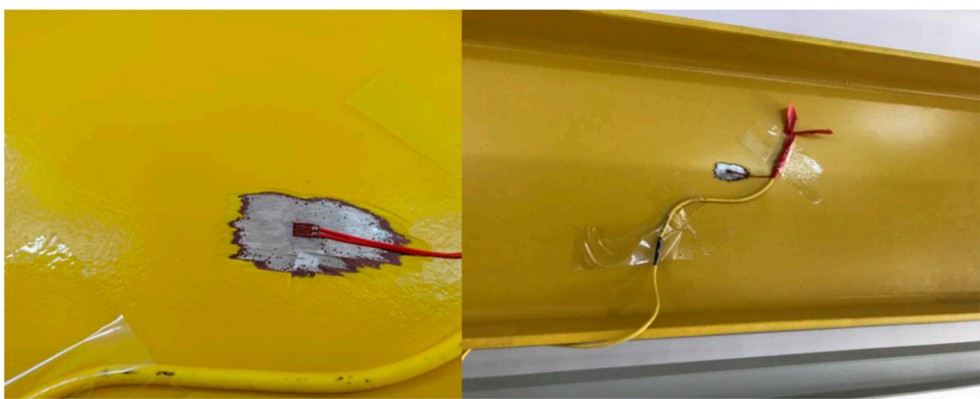

**Figure 12.** The stress–strain gauges for the crane.

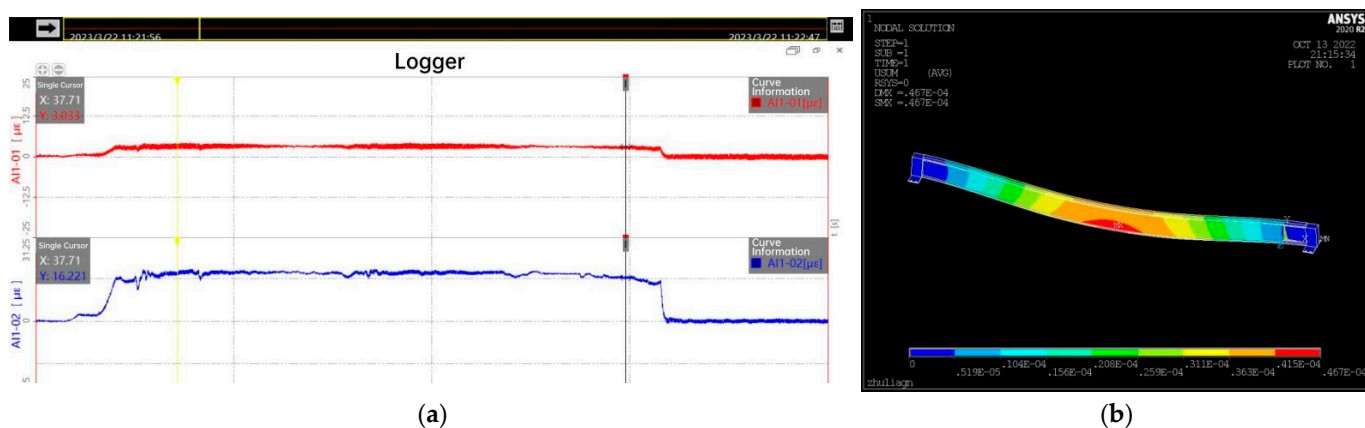

(**a**)          (**b**)

**Figure 13.** The stress analysis of the main beam. (**a**) The results of stress-time curve analysis. (**b**) The results of finite-element analysis.

The predictive maintenance system of the overhead crane based on digital twins could display the operation status of the crane in real time and display the status of the main girder, e.g., the finite element analysis result and the fault code, as shown in Figure 14.

From the log records of the serial communication experiment data, the following observations can be made: When the overhead crane rose, its load-bearing stress value would reach the peak in 1.5 to 3 s, and it rapidly increased from $-5 \times 10^{-4}$ MPa to 0 MPa, then to $20$–$25 \times 10^{-4}$ MPa, then remained at the same level. When the cart or trolley of the overhead crane moved in the horizontal direction to change its position, the stress value remained unchanged, and there were no obvious fluctuations in this period. When the overhead crane performed a descending operation, its load-bearing stress would fall back to about 0 MPa again in 3 to 5 s.

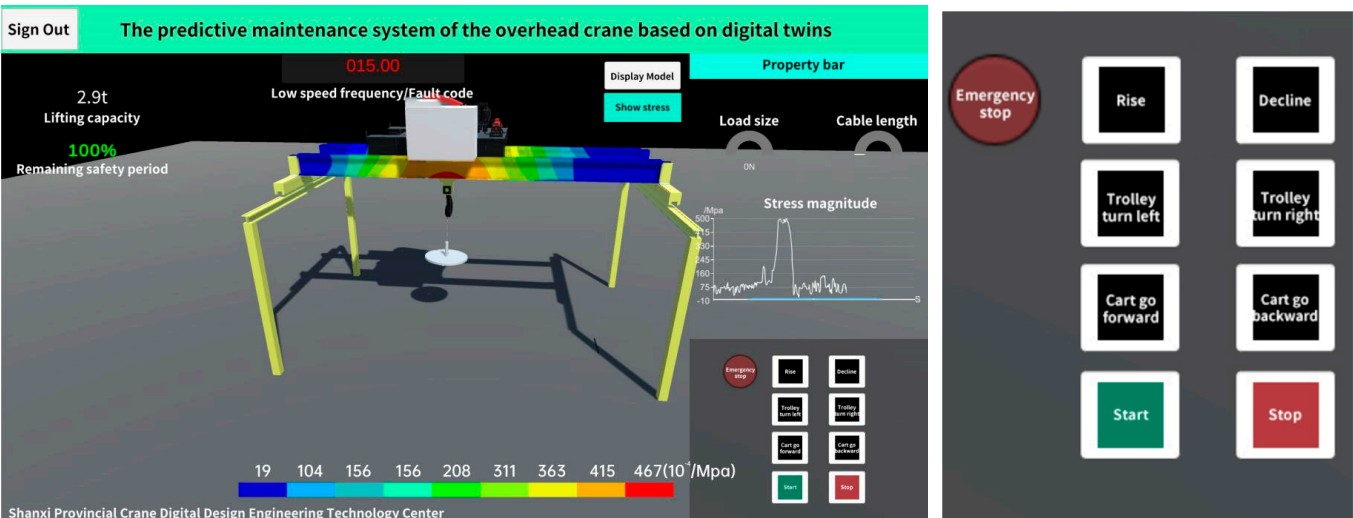

**Figure 14.** Display of stress and strain.

Meanwhile, the digital twin program rendered corresponding color changes according to the actual movement instructions of the crane. The instrument panel on the right side of the simulation software highlighted the weight of the crane cargo and the length of the cable in real time. The true stress value at each time point was recorded on a linear graph, and the linear image had an almost consistent variation trend with the observed stress value of the overhead crane.

During this experiment, when the crane had abnormal data and faults, the observed crane status was in line with the data recorded via serial communication and the prompt of the simulation software, which was consistent with the expected design judgment.

### 5.4. Life Prediction Results

Through experiments and simulations, 145 data sets related to stress and crane life were obtained. Based on these 145 data sets, a BP neural network prediction model was established in a supervised learning mode, i.e., the establishment of the model was completed via dividing the data sets into training sets and test sets. Through multiple experiments, in the 145 data sets, when the ratio of the training sets to the test sets was 25:4 (125 training sets and 20 test sets), the training performance was the best. The error comparison of the prediction data before and after optimization, executed through verification with the test sets, is shown in Figure 15.

Figure 16 is a schematic diagram of the comparison between the predicted life values and the measured values of the 20 test sets. As shown in Figure 16, the predicted life values were close to the measured ones, and there were small errors between them. Thus, this prediction model had a better prediction effect.

It can be seen from the results in Figure 16 that the prediction model achieved a good prediction effect. However, to reflect the overall prediction effect more intuitively, the coefficient of determination was taken as an evaluation indicator to evaluate the quality of the model prediction. Its calculation is shown in Equations (4)–(6). After calculation, the coefficient of determination, $R^2$, was 0.83198, indicating that the overall prediction model had better prediction performance and higher accuracy. Thus, the life prediction in the crane digital twin system met the actual engineering requirements.

$$SS_{res} = \Sigma(y_i - f_i)^2 \tag{4}$$

$$SS_{tot} = \Sigma(y_i - \overline{y})^2 \tag{5}$$

$$R^2 = 1 - \frac{SS_{res}}{SS_{tot}} \tag{6}$$

**Figure 15.** The prediction errors before and after optimization.

**Figure 16.** Comparison of the predicted life and measured life.



## 6. Conclusions

In this study, Unity3D was utilized to develop a digital twin program of an overhead crane. From crane modeling to software design simulation, the crane operation simulation was realized via adopting digital approaches. Consequently, the physical characteristics of the crane in the working process were fed back in a real, timely, and intuitive manner. The stress method was used to predict the remaining life of the bridge crane's main beam, obtain its stress load and time–stress curve, and derive the calculation formula for its remaining fatigue life.

The keys to overhead-crane digital visualization technology are to collect the all-around data of the machine and to conduct data interaction with a simulation program through serial communication. Through multiple experiments and joint debugging tests, various physical characteristics of the overhead crane could be fully perceived. Meanwhile, according to physical characteristics, programs were written to realize functions, and the logic algorithm was optimized to improve the program execution efficiency. For the simulation software, this study used Modbus CRC16 to verify serial port data with the crane; this software can respond timely and rapidly to send and receive instruction data and operating status data in the byte stream mode.

This study realized the remote delivery of work movement instructions to an overhead crane through serial communication. In addition, the physical data of the crane could be transmitted to the simulation interface synchronously, thus ensuring the real time and validity of the data.

**Author Contributions:** Conceptualization, P.G. and Y.Z.; methodology, Y.Z.; software, P.G.; validation, P.G., Y.Z., and M.L.; formal analysis, P.G.; investigation, P.G.; resources, M.L.; data curation, P.G.; writing—original draft preparation, P.G.; writing—review and editing, P.G.; visualization, M.L.; supervision, Z.W.; project administration, Z.W.; funding acquisition, Y.Z. All authors have read and agreed to the published version of the manuscript.

**Funding:** This research was funded by the Shanxi Provincial Key Research (International Cooperation) Project Application Agreement for Cooperation (201903D421015).

**Data Availability Statement:** Not applicable.

**Acknowledgments:** The authors would like to thank the Shanxi Provincial Key Research (International Cooperation) Project Application Agreement for Cooperation for its funding.

**Conflicts of Interest:** The authors declare no conflict of interest.

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
