# Peer review of "Prediction System for Overhead Cranes Based on Digital Twin Technology"

_applsci, doi:10.3390/app13084696_

Round 1

Reviewer 1 Report

The paper has a very interesting topic and presents an interesting practical demonstration. Unfortunately, the scientific contributions of the paper are unclear. In the introduction section, it is stated that the paper is about prediction with digital twins of the crane, but the prediction algorithms or application on the maintenance is not clearly presented in the rest of the paper. Also, in the abstract, it is unclear for which purposes this prediction system is used. What does it predict?

What are the main contributions of the paper? They could be stated at the end of the introduction to make the paper clearer. This could also help the reader to understand the scientific contributions of the paper.

The paper is sometimes very hard to understand, and it is difficult to follow the ”storyline”; The paper requires significant work on the text quality.

The title could be shortened, for example, to “Prediction System for Overhead Cranes Based on Digital Twin Technology”. Open all abbreviations such as “TTL”, EIA”, and “MVVM

What is com terminal transfer method?

With some figures, the figure quality seems poor, a higher resolution should be used.

There is a lot of code on the paper. You should only include relevant codes. Some basic code for implementing UI is not very interesting for the audience. I encourage you to load your code into GitHub and publish it as open source.

I guess the crane is not weighing 2,9 tons, but it is capable of lifting 2.9 tons. (Line 418)

Overall, it is hard to understand what is the point of the paper: is it predicting the crane maintenance need, is it calculating the stresses of the crane, or is it the system as whole?

I see potential in the paper, if the message of the paper is sharpened including the scientific contributions, the text quality is improved, and the methods are presented more clearly.

Author Response

Point 1: The paper has a very interesting topic and presents an interesting practical demonstration. Unfortunately, the scientific contributions of the paper are unclear. In the introduction section, it is stated that the paper is about prediction with digital twins of the crane, but the prediction algorithms or application on the maintenance is not clearly presented in the rest of the paper. Also, in the abstract, it is unclear for which purposes this prediction system is used. What does it predict?

Response 1: Thank you very much for your constructive comments.This paper focuses on predicting the remaining life cycle of a crane using the stress variation in the main beam. The content has been added in the abstract, introduction section, and section 4 of the paper.

Point 2: What are the main contributions of the paper? They could be stated at the end of the introduction to make the paper clearer. This could also help the reader to understand the scientific contributions of the paper.

Response 2: The contributions of this paper have been added at the end of the introduction section and summarized in the conclusion section.

Point 3: The paper is sometimes very hard to understand, and it is difficult to follow the ”storyline”; The paper requires significant work on the text quality.

Response 3:Thank you very much for your valuable comments.I have improved the "storyline", including adding section 4 and additional experimental content. In terms of the quality of the writing language, the manuscript has been revised by a professional polishing agency, and the polishing certificate is included in the end of the manuscript.

Point 4: The title could be shortened, for example, to “Prediction System for Overhead Cranes Based on Digital Twin Technology”. Open all abbreviations such as “TTL”, “EIA”, and “MVVM”.

Response 4: The title of this paper is a bit long. I have revised it according to your suggestions, and I thank you again for your advice.The abbreviations in the paper have all been explained.

Point 5: What is com terminal transfer method?

Response 5: The com terminal transmission method is to control the communication with the crane through a computer com terminal connected to a transmitter. This issue is explained in section 5.2.

Point 6: With some figures, the figure quality seems poor, a higher resolution should be used.

Response 6: The unclear pictures have been changed, e.g., Fig. 3, Fig. 7, and Fig. 11.

Point 7: There is a lot of code on the paper. You should only include relevant codes. Some basic code for implementing UI is not very interesting for the audience. I encourage you to load your code into GitHub and publish it as open source.

Response 7: Your suggestion is valuable. A lot of basic code should not have been put in the paper and has been removed entirely.

Point 8: I guess the crane is not weighing 2,9 tons, but it is capable of lifting 2.9 tons. (Line 418)

Response 8: I am very sorry for my mistake. Your suggestion is valuable, and I have revised the expression.

Point 9: Overall, it is hard to understand what is the point of the paper: is it predicting the crane maintenance need, is it calculating the stresses of the crane, or is it the system as whole?

Response 9: Please forgive me for not highlighting the main point of the paper, which is that the remaining life cycle of the crane can be predicted by calculating the stresses of the crane in the digital twin system. The details have been added to the paper.

Reviewer 2 Report

Review:

"In the manuscript, the authors take up a very interesting issue, which is the prediction system of transport devices, for which they use the Digital Twin technology. The approach is original and the topic is current. Despite the interesting topic, the article unfortunately has several issues that need to be developed or corrected before final publication.

Remarks: 1. The literature review is not exhaustive. I adding a few/dozen works to fully illustrate the novelty of the proposed method. In addition, the novelty of work is not properly listed, making it hard to guess what the most important original achievement is.

2. Moderate English changes required. Authors should read the article carefully again because it contains occasional linguistic or stylistic errors. Also, I would suggest using proper scientific notation (e.g. line 442, line 447).

3. The presented methods should be better described, and the discussion of the results should be extended. Instead of inserting whole codes (which in some cases are simple, easily accessible by others), the authors should focus on describing the methods (crane stresses, system prediction etc.)

4. The quality of some drawings must be improved - figure resolution is too small."

Author Response

Point 1: The literature review is not exhaustive. I adding a few/dozen works to fully illustrate the novelty of the proposed method. In addition, the novelty of work is not properly listed, making it hard to guess what the most important original achievement is.

Response 1: Thank you very much for your constructive comments.I have added the current state of research on crane life prediction to the literature review section, which is covered by section 2.3.The most important original achievement is: this paper uses a digital twin system to achieve synchronous motion of the crane equipment and the model and collects data during the crane motion to predict the remaining life cycle of the crane using the stress change data of the main beam. Four achievements have been added at the end of the introduction section and summarized in the conclusion section.

Point 2:  Moderate English changes required. Authors should read the article carefully again because it contains occasional linguistic or stylistic errors. Also, I would suggest using proper scientific notation (e.g. line 442, line 447).

Response 2: I am very sorry for my mistakes. The manuscript has been revised by a professional polishing agency to improve the quality of the writing, and the polishing certificate is included in the end of the manuscript.The figures in the paper have been corrected by adding scientific notations.

Point 3:  The presented methods should be better described, and the discussion of the results should be extended. Instead of inserting whole codes (which in some cases are simple, easily accessible by others), the authors should focus on describing the methods (crane stresses, system prediction etc.)

Response 3: A description of the proposed lifetime prediction method has been added, and the experimental findings have been further analyzed.A lot of code should not be placed in the paper, and I have removed all of it. The introduction of the crane life prediction algorithm has been added in section 4; the stress experimental link of the main beam of the crane has been added in section 5.

Point 4:  The quality of some drawings must be improved - figure resolution is too small.

Response 4: The unclear pictures have been changed, e.g., Fig. 3, Fig. 7, and Fig. 11.
